# Galectin-1 and platelet factor 4 (CXCL4) induce complementary platelet responses in vitro

**Annemiek Dickhout**[1☯], **Bibian M. E. Tullemans**[1☯], **Johan W. M. Heemskerk**[1], **Victor L. J. L. Thijssen**[2]*, **Marijke J. E. Kuijpers**[1], **Rory R. Koenen**[1,3]*

**1** Department of Biochemistry, Cardiovascular Research Institute Maastricht (CARIM), Maastricht University, Maastricht, The Netherlands, **2** Amsterdam UMC, location VUmc, Medical Oncology & Radiation Oncology, Amsterdam, The Netherlands, **3** Institute for Cardiovascular Prevention (IPEK), LMU Munich, Munich, Germany

☯ These authors contributed equally to this work.
* r.koenen@maastrichtuniversity.nl (RRK); v.thijssen@amsterdamumc.nl (VLJLT)

**Data Availability Statement:** All relevant data are within the manuscript and its Supporting information files.

## Abstract

Galectin-1 (gal-1) is a carbohydrate-binding lectin with important functions in angiogenesis, immune response, hemostasis and inflammation. Comparable functions are exerted by platelet factor 4 (CXCL4), a chemokine stored in the α-granules of platelets. Previously, gal-1 was found to activate platelets through integrin $\alpha_{IIb}\beta_3$. Both gal-1 and CXCL4 have high affinities for polysaccharides, and thus may mutually influence their functions. The aim of this study was to investigate a possible synergism of gal-1 and CXCL4 in platelet activation. Platelets were treated with increasing concentrations of gal-1, CXCL4 or both, and aggregation, integrin activation, P-selectin and phosphatidyl serine (PS) exposure were determined by light transmission aggregometry and by flow cytometry. To investigate the influence of cell surface sialic acid, platelets were treated with neuraminidase prior to stimulation. Gal-1 and CXCL4 were found to colocalize on the platelet surface. Stimulation with gal-1 led to integrin $\alpha_{IIb}\beta_3$ activation and to robust platelet aggregation, while CXCL4 weakly triggered aggregation and primarily induced P-selectin expression. Co-incubation of gal-1 and CXCL4 potentiated platelet aggregation compared with gal-1 alone. Whereas neither gal-1 and CXCL4 induced PS-exposure on platelets, prior removal of surface sialic acid strongly potentiated PS exposure. In addition, neuraminidase treatment increased the binding of gal-1 to platelets and lowered the activation threshold for gal-1. However, CXCL4 did not affect binding of gal-1 to platelets. Taken together, stimulation of platelets with gal-1 and CXCL4 led to distinct and complementary activation profiles, with additive rather than synergistic effects.

## Introduction

Galectins are an evolutionary conserved family of carbohydrate-binding lectins, with binding specificity for $\beta$-galactoside-containing glycans. Although structurally diverse, all members of the galectin family contain either one or two so-called carbohydrate recognition

**Funding:** This study was supported by Netherlands Foundation for Scientific Research (ZonMW) (www.zonmw.nl) in the form of a grant awarded to RRK (VIDI 016.126.358) and the Landsteiner Foundation for Blood Transfusion Research (LSBR) (www.lsbr.nl) in the form of a grant awarded to RRK (1638). CARIM School for Cardiovascular Diseases, Universiteit Maastricht financed the salary of AD (1198N), and the Maastricht Thrombosis Expertise Centre provided infrastructural support to MJEK. The funders had no role in study design, data collection and analysis, decision to publish, or preparation of the manuscript.

**Competing interests:** The authors have declared that no competing interests exist.

domains (CRD), which are highly conserved and specific for each lectin. Homodimerization or multimerization of the CRDs increases the binding valency and facilitates complex interaction with glycoconjugates, like glycoproteins and glycolipids. This allows galectins to contribute to e.g. cell-matrix and cell-cell interactions as well cell signalling. Galectin-1 (gal-1) is the first family member to be described and one of the best studied galectins. The CRD of gal-1 (±14 kD) has a binding affinity for type I and type II N-acetyllactosamine disaccharides (LacNAc) and the protein exists as a non-covalent homodimer [1, 2]. It is expressed by different cell types and plays distinct roles in different cellular functions. For example, gal-1 is expressed by endothelial cells and was shown to have a key role in angiogenesis, as knockdown in zebrafish results in vascular defects and in *gal-1*$^{-/-}$ mice, tumor growth is delayed [3, 4]. In addition, gal-1 is an important regulator of the immune response and inflammation, by controlling the recruitment, survival and function of distinct immune cell populations [5–7]. Currently, gal-1 is also emerging as a regulator of platelet function. Proteomics and immunoblotting have shown that gal-1 is present in human [8] and mouse platelets [9]. Moreover, the presence of gal-1 mRNA in platelets and megakaryocytes is also described in the blueprint database (https://blueprint.haem.cam.ac.uk/bloodatlas). It was previously shown by Pacienza and colleagues that gal-1 has the ability to induce platelet aggregation [10]. This effect is completely abolished when gal-1 is incubated with lactose, indicating that binding is dependent on specific carbohydrate ligands present on platelets. The authors narrowed down possible gal-1 binding targets on platelets to the integrin $\alpha_{IIb}\beta_3$, which is not only important for platelet aggregation but also potentiates platelet responses through outside-in signaling [9, 11]. Moreover, *gal-1*$^{-/-}$ mice have a prolonged bleeding time with a normal platelet count [9]. In addition to integrin $\alpha_{IIb}\beta_3$, other binding partners for gal-1 have also been described that are heavily glycosylated, being von Willebrand factor [12] and factor VIII [13]. Gal-1 (and also gal-3) have been shown to modulate the activity of these proteins. Thus, gal-1 appears to have a function as a regulator of (tumor) angiogenesis, hemostasis and inflammatory responses. A structurally distinct protein acting in the same biological processes is platelet factor 4 (PF4, CXCL4). CXCL4 is present in the $\alpha$-granules of platelets in amounts that constitute up to 2% platelet protein mass. After platelet activation, local concentrations can increase over a 100-fold and serum levels can reach up to 1.9 $\mu$M [14]. On a structural level, CXCL4 belongs to the family of chemokines, i.e. small chemotactic cytokines, which has more than 50 members. However, whereas other chemokines induce intracellular calcium mobilization, chemotaxis and leukocyte recruitment in picomolar to nanomolar concentration ranges, CXCL4 does not appear to act on cells in these low concentrations. Nevertheless, CXCL4 mediates several physiologic processes e.g. by inhibiting (tumor) angiogenesis and by promoting inflammation [15, 16]. With regard to hemostasis, *pf4*$^{-/-}$ mice do not show impaired bleeding, but have impaired thrombus formation in an in vivo thrombosis model [17]. CXCL4 has also been shown to modulate the activation of protein C [18, 19]. These studies indicate that CXCL4 also plays a role in thrombus formation and coagulation. Some of these functions are thought to be mediated by interactions of CXCL4 with other biomolecules, e.g. negatively charged polysaccharides (glycosaminoglycans) [20, 21], nucleic acids [16], angiogenic factors (e.g. FGF2) [22], integrins [23] or other chemokines (e.g. CCL5) [24]. Of note, the interaction of CXCL4 with polyanions is crucially involved in the pathophysiology of heparin-induced thrombocytopenia (HIT), a threatening (auto-)immune reaction with a strongly increased risk for thrombosis [20, 25]. Given the high affinity of CXCL4 for polysaccharides and the similarity of exerted functions, we aimed to investigate whether gal-1 and CXCL4 can influence each other's activities during platelet activation.

## Materials and methods

### Blood collection and handling

Human blood was collected from healthy volunteers by venepuncture using a vacutainer tube containing 3.2% trisodium citrate, after obtaining informed consent in accordance with the Helskini declaration. The study was approved by the medical ethics committee of the Maastricht University Medical Center+ (MUMC+). The first 3 mL of blood were discarded, and washed platelets were prepared as described before [26]. Briefly, blood was centrifuged for 15 minutes at 240g, to obtain platelet-rich plasma (PRP). PRP was then supplemented with 1:10 acidic citrate dextrose (ACD, 80 mM trisodium citrate, 52 mM citric acid and 180 mM glucose) and centrifuged for 2 minutes at 2230g. The platelets were then resuspended in Hepes buffer pH 6.6 (10 mM Hepes, 136 mM NaCl, 2.7 mM KCl, 2 mM MgCl2, 5 mM glucose and 0.1% BSA) and supplemented with 1:15 ACD and 1 U/mL apyrase (Sigma-Aldrich A6535, St. Louis, MO, USA). After another centrifugation at 2230g, the platelets were finally placed in Hepes buffer pH 7.5. Platelet count was determined using a Sysmex XP300 hematology analyser (Kōbe, Japan).

### Confocal microscopy

Washed platelets ($50 \times 10^9$ platelets/L) were allowed to spread over poly-L-lysine (Sigma-Aldrich P4707)-coated 12 wells plate for 30 minutes at 37°C, and next incubated with recombinant 3 μM Oregon Green-labelled gal-1. Platelets were then fixed for 20 minutes at room temperature with 1% paraformaldehyde in PBS. CXCL4 was detected using a rabbit polyclonal antibody against CXCL4 (2μg/ml, Peprotech 500-P05, Rocky Hill, NJ, USA), and visualization was performed using donkey-anti-rabbit Alexa Fluor 647 (5 μg/mL, Thermo Fisher Scientific A-31573, Waltham, MA, USA). Recombinant gal-1 was expressed in E. coli and purified essentially as described [27].

### ELISA analysis of platelet releasates

Washed platelets were incubated with 1, 3 or 6 μM gal-1, or type I collagen (5 μg/mL "Horm"—Takeda diagnostics, Tokio, Japan) for 30 minutes at 37°C under stirring conditions, and subsequently centrifugated for 2 minutes at 2230g. Supernatants were frozen until further analysis. Sandwich ELISA analysis was performed using a human CXCL4/CXCL4 DuoSet ELISA (R&D systems, Minneapolis, MN, USA) and a serotonin ELISA kit (Abnova, Taipeh, Taiwan) according to the manufacturers' instructions.

### Flow cytometry

Washed platelets ($100 \times 10^9$ platelets/L) were incubated for 1 hour at 37°C without or with $\alpha$2-3–$\alpha$2-6-neuraminidase from C. perfringens (200 mU/$10^9$ platelets, Abnova P5289), and next with gal-1 (0.1-6 μM), CXCL4 (1-12 μM), and/or Tirofiban (1 μg/mL—Merck, Darmstadt, Germany) and 1 μg/mL cross-linked collagen-related peptide (CRP-XL, obtained from the Dept. Biochemistry, Cambridge University) for 15 minutes. Some experiments were carried out in the presence of 10μg/mL of heparin (LeoPharma, Ballerup, Denmark) in order to block endogenously released CXCL4 [28]. Integrin $\alpha_{IIb}\beta_3$ activation was determined using a FITC-conjugated PAC1 mAb (1:10, BD Biosciences #340507, Heidelberg, Germany), P-Selectin expression was determined using a FITC-conjugated CD62-P mAb (1:10, Beckman Coulter #65050, Brea, CA, USA) and phosphatidylserine (PS) exposure was determined using a FITC-conjugated annexin A5 (1 μg/mL, PharmaTarget, Maastricht, The Netherlands) in the presence of 5 mM calcium chloride. The data was expressed as percentage fluorescence-positive platelets of the total platelet count, with the marker set at approx. 0.67 times the fluorescence

of quiescent platelets. Binding experiments were carried out by adding increasing concentrations of FITC-labeled gal-1 (0-1$\mu$M), without or with pre-incubation with equimolar amounts of CXCL4 for 60 min, prior to addition to washed platelets, without or with neuraminidase treatment. The $\alpha_{\text{IIb}}\beta_3$ inhibitors eptifibatide (10$\mu$M, GlaxoSmithKline, Brentford, UK), tirofiban (1$\mu$g/mL, Merck), or abciximab (50$\mu$g/mL, Johnsson&Johnsson, New Brunswick, NJ, USA) were incubated with the platelets prior to addition of labeled gal-1 or gal-1/CXCL4. Flow cytometry was performed with standardized methods using an Accuri™ flow cytometer (BD Biosciences) and accompanying software.

## Light transmission aggregometry

Washed platelets (250x10$^9$ platelets/L) were incubated for 1 hour at 37˚C with or without the presence of neuraminidase (200 mU/10$^9$ platelets). Aggregation was measured using a Chrono-log optical aggregometer (Havertown, PA, USA) under constant stirring at 37˚C, after the addition of collagen (5 $\mu$g/mL), gal-1 (0.25 − 6 $\mu$M) or recombinant human CXCL4 (1-12 $\mu$M) as indicated. In some experiments, PRP was stimulated with collagen (5 $\mu$g/mL) or gal-1 (1 − 6 $\mu$M). For stimulation of platelets with the combination of gal-1 and CXCL4, the proteins were incubated together for 1 hour (at 4˚C) in order to allow any potential interactions to take place.

## Surface plasmon resonance

Binding studies of CXCL4 to heparin were carried out on a BiaCore T100 (GE Healthcare Life Sciences, Chicago, IL) using a CMD200M chip (XanTec bioanalytics GmbH, Düsseldorf, Germany). First, streptavidin (100 $\mu$g/mL, Sigma) in 10 mM sodium acetate, pH4.5 was immobilized to approx. 6000 resonance units (RU). Then, biotinylated heparin (1$\mu$g/ml) Sigma-Aldrich B9806) was perfused at 5 $\mu$L/min for 300 sec to achieve approx. 500 RU immobilization. CXCL4 was perfused in varying concentrations (0-50nM) in 10 mM Hepes, 150 mM NaCl pH7.4 and 0.002% tween-20 (HBS-T) in the absence or presence of 500nM gal-1 at 10 $\mu$L/min for 180 sec followed by 30 sec dissociation. Complete regeneration was achieved by 2 sequential wash steps (1) heparin (50$\mu$g/mL, LeoPharma) in HBS-T (180 sec) followed by (2) 0.1 M Tris, 2 M NaCl pH9.5 for 180 sec. A flow cell containing only streptavidin served as background control. Next, CXCL4 (50nM) was perfused for 60 sec to obtain a stable CXCL4/heparin baseline. Then, the heparin-induced thrombocytopenia model antibody KKO (Thermo Fisher Scientific) [20, 25] was perfused in HBS-T from 0-125 nM in the absence or presence of 500nM gal-1 at 10 $\mu$L/min for 900 sec followed by 300 sec dissociation. The flow cell was regenerated as described above. Binding traces were recorded as independent triplicates.

## Statistical analysis

Data is presented as mean ± SD, unless stated otherwise. Statistical analysis was performed with Graphpad Prism v 8.3.1 (San Diego, USA), using one way non-parametric analysis of variance (Kruskal Wallis) and Dunn's post hoc analysis. Stimulations of gal-1 or CXCL4 in presence or absence of neuraminidase were compared to control. Differences were considered statistically significant at P < 0.05 (*), P < 0.01 (**) or P < 0.001 (***).

# Results

## Galectin-1 causes release of $\alpha$-granules and colocalizes with CXCL4 on the platelet surface

As it has been described that gal-1 activates platelets and induces P-selectin expression [9], we investigated whether gal-1-mediated platelet activation can induce CXCL4 release from $\alpha$-

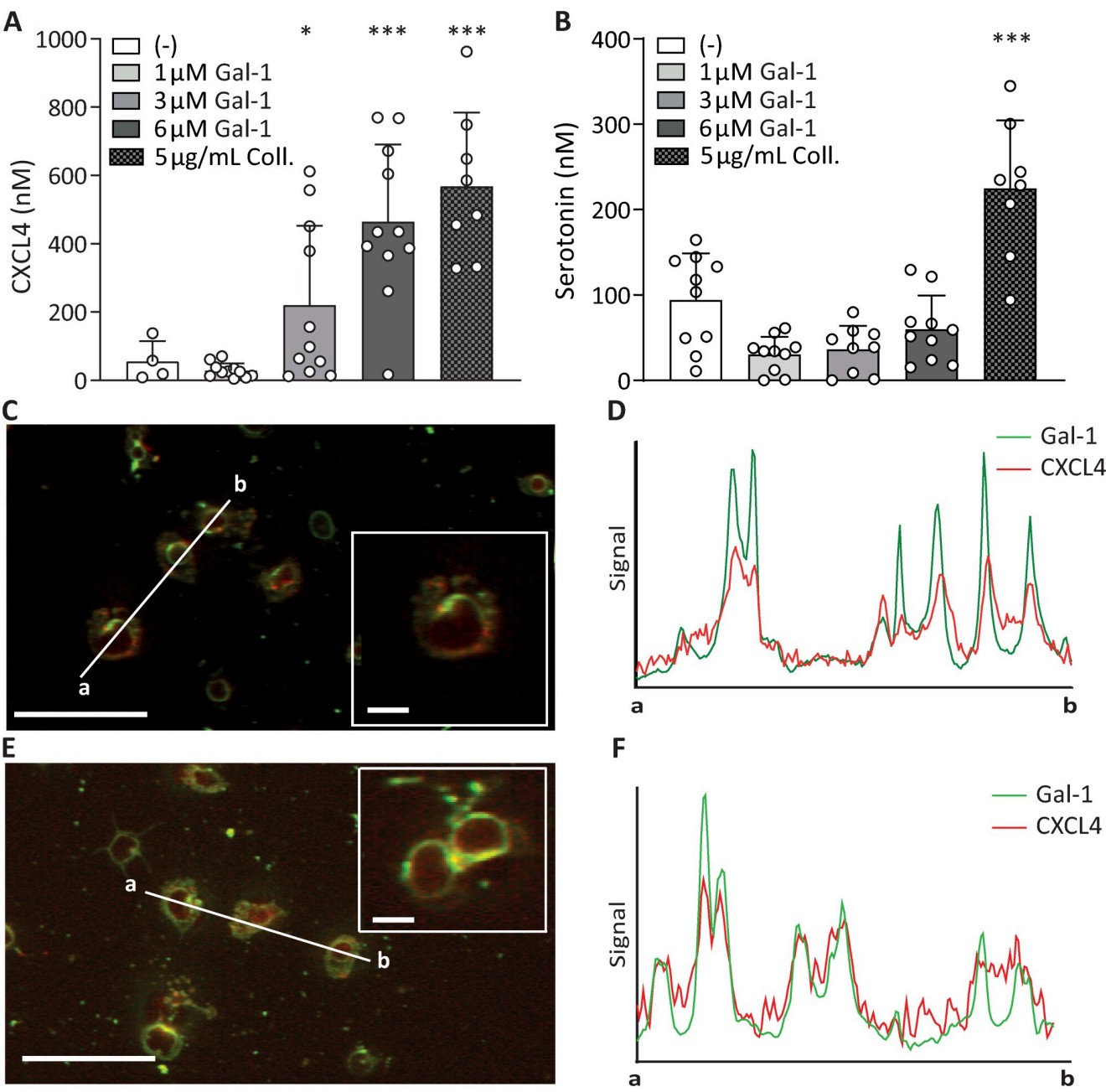

**Fig 1. Galectin-1 binds to platelets, co-localizes to CXCL4 and causes CXCL4 release.** Absolute levels of CXCL4 (A) and serotonin (B) released from platelets activated with gal-1 (1, 3 or 6 μM), or 5 μg/mL collagen as a control α-granule release agent. (C,E) Confocal microscopy of spread platelets on poly-L-lysine. Platelets were then incubated with OG-gal-1 (green), then fixed and stained with a primary antibody against CXCL4, and AF-647 labeled secondary antibody (red). Inset: enlarged section of image. (D,F) Intensity profile through 3 adjacent platelets, indicated by line a-b in C and E, showing gal-1 levels (green) and CXCL4 levels (red).*p < 0.05, ***p < 0.001 as compared to control (Kruskal Wallis/Dunn's test). Scale bar: 25 μm, inset: 5 μm.

granules. Gal-1 (1, 3, and 6μM) was found to dose-dependently trigger CXCL4 release from platelets ([Fig 1A]). The addition of gal-1 at these concentrations resulted in released levels of CXCL4 similar to those observed for collagen, an established stimulus for CXCL4 release from α-granules [29]. The release of serotonin, stored in dense granules, was not dose-dependently induced by gal-1 ([Fig 1B]). To further characterize the functional relationship between gal-1 and

CXCL4 in platelet function, we investigated the binding of gal-1 and CXCL4 to platelets by confocal microscopy. Platelets were allowed to spread on a poly-L-lysine-coated surface and were incubated with fluorescently-labelled gal-1, and subsequently fixed and co-stained for CXCL4. Staining of gal-1 was found to colocalize with that of released CXCL4, since a cross-sectional intensity profile through three platelets showed a clear correlation between the green and red markers (Fig 1C–1F). Thus, gal-1 induces the release of CXCL4 from platelets, which can be found both in the supernatant and on the platelet surface, where it colocalizes with gal-1.

## Galectin-1 and CXCL4 activate platelets through different pathways

Since gal-1 has the ability to induce $\alpha$-granule release and colocalizes with CXCL4, we hypothesized that these proteins activate platelets through similar pathways. To test this, $\alpha_{IIb}\beta_3$ activation, P-selectin expression and phosphatidylserine (PS)-exposure were measured after treatment of washed platelets with CRP-XL, and either gal-1 (Fig 2) or CXCL4 (Fig 3) using flow cytometry. Interestingly, gal-1 treatment induced a robust activation of integrin $\alpha_{IIb}\beta_3$ (Fig 2A and 2E) without surface P-selectin presentation (Fig 2B and 2F), whereas CXCL4 treatment induced an opposite response with low levels of active integrin $\alpha_{IIb}\beta_3$ (Fig 3A and 3E) and high levels of surface P-selectin (Fig 3B and 3F). Neither gal-1 nor CXCL4 treatment induced notable PS-exposure (Figs 2C and 2G and 3C and 3G). The response of platelets to gal-1 was not altered in the presence of heparin (S1 Fig), which was added to block endogenously released CXCL4 (cf. Fig 1A). Since gal-1 and CXCL4 treatment induced integrin $\alpha_{IIb}\beta_3$ activation or P-selectin presentation, respectively, it was investigated whether these proteins can also induce platelet aggregation. Indeed, exposure of washed platelets to either protein induced a dose-dependent aggregation response up to 80% with 6 $\mu$M gal-1 (Fig 2D and 2H) and up to 50% with 12 $\mu$M CXCL4 after 15 minutes (Fig 3D and 3H). Of note, the aggregation levels did not further increase after this time point for up to 24 minutes measured. Interestingly, in contrast to the positive control collagen, platelet aggregation in response to gal-1 and CXCL4 displayed a biphasic pattern, with a lag time of 5 minutes for gal-1 and 11-12 minutes

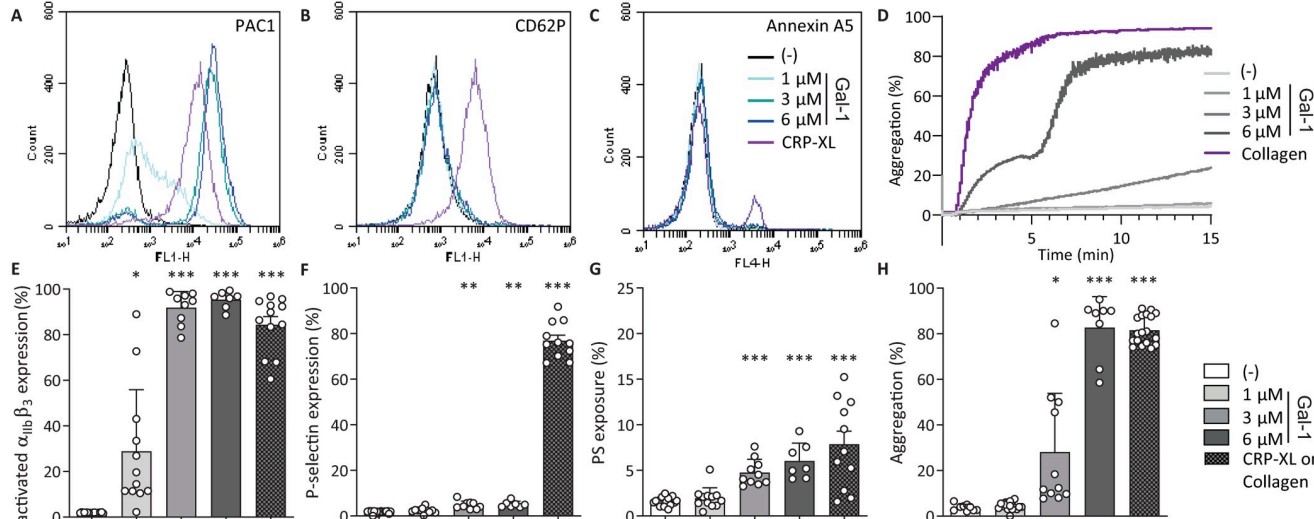

**Fig 2. Galectin-1 induces different platelet responses.** Flow cytometry measurements of platelets, assessing activation of integrin $\alpha_{IIb}\beta_3$ and P-selectin using fluorescently labelled PAC-1 (A, E), anti-CD62-P (B, F), and annexin A5 (C, G), respectively. Representative histograms represent washed platelets incubated with different concentrations of gal-1 (A-C) or 1 µg/mL CRP-XL for 15 minutes and bar graphs represent the percentage of platelet $\alpha_{IIb}\beta_3$ activation (E), P-selectin expression (F) and PS-exposure (G) by gal-1. Light transmission aggregometry of washed platelets (D) and quantification (H), incubated with different concentrations of gal-1 or 5 µg/mL collagen for 15 minutes. Bars represent mean ± SD (n = 3-5 measured in duplo). $^*$p < 0.05, $^{**}$p < 0.01, $^{***}$p < 0.001 as compared to control (Kruskal Wallis/Dunn's test).

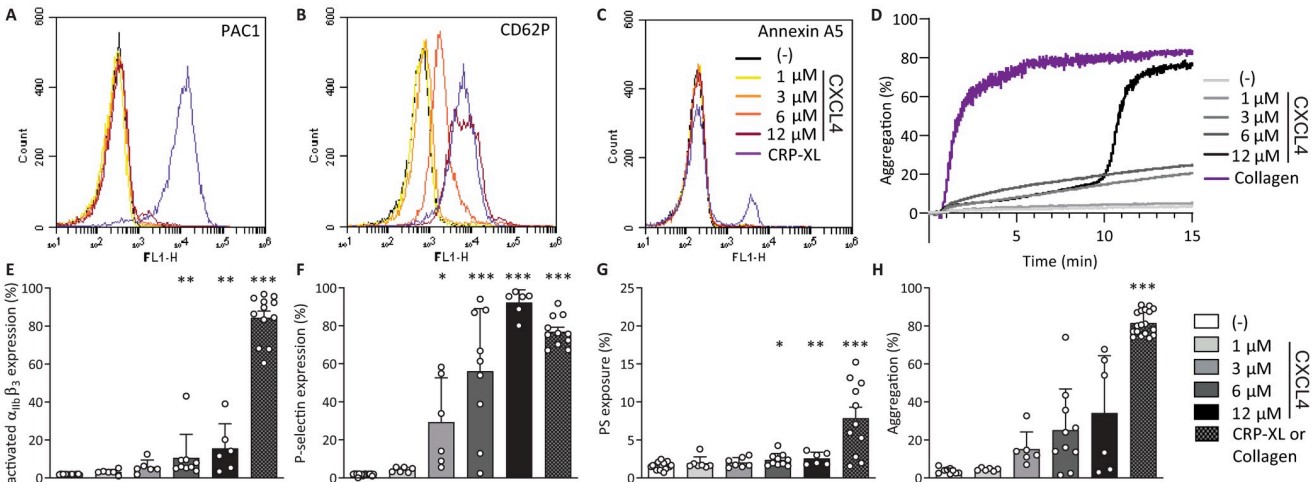

**Fig 3. CXCL4 induces different platelet responses.** Flow cytometry measurements of platelets, assessing activation of integrin $\alpha_{IIb}\beta_3$ and P-selectin using fluorescently labelled PAC-1 (A), anti-CD62-P (B), and annexin A5 (C), respectively. Representative histograms represent washed platelets incubated with different concentrations of CXCL4 (A-C) or 1 µg/mL CRP-XL for 15 minutes and bar graphs showing the percentage of platelet $\alpha_{IIb}\beta_3$ activation (E), P-selectin expression (F) and PS-exposure (G) by CXCL4. Light transmission aggregometry of washed platelets (D) and quantification (H), incubated with different concentrations of CXCL4 or 5 µg/mL collagen for 15 minutes. Bars represent mean ± SD (n = 3-5 measured in duplo). *p < 0.05, **p < 0.01, ***p < 0.001 as compared to control (Kruskal Wallis/Dunn's test).

for CXCL4, pointing towards secondary secretion-dependent effects. Unlike collagen, gal-1 did not induce aggregation of platelets in PRP (S2 Fig).

## Galectin-1 and CXCL4 have additive activation effects on platelets

After determining the separate effects of gal-1 and CXCL4 on platelet activation and aggregation, the proteins were co-incubated prior to addition to platelets, in order to explore whether they had mutually influencing effects, e.g. resulting in an altered platelet response. Co-incubation of 1 µM gal-1, which triggered intermediate platelet responses in previous experiments, with increasing concentrations of CXCL4 resulted in a dose-dependent increase of integrin $\alpha_{IIb}\beta_3$ and P-selectin responses (Fig 4A and 4B). The activation of integrin $\alpha_{IIb}\beta_3$ increased dose-dependently after activation with 1 µM gal-1 and increasing doses of CXCL4. P-selectin expression was induced with gal-1 and CXCL4 to a similar extent as with CXCL4 alone (Figs 3F and 4B). Thus, gal-1 did not affect the function of CXCL4 in the induction of P-selectin secretion. There was no significant increase of PS exposure after co-stimulation with gal-1 and CXCL4 (Fig 4C). Platelet aggregation did show a dose-response after stimulation with 1 µM gal-1 preincubated with increasing doses of CXCL4 (Fig 4D and 4E). Although platelet aggregation appeared to be induced in an all-or-none fashion under these conditions, a significantly higher aggregation was observed with 1 µM gal-1 and 6 µM CXCL4, compared with 1 µM gal-1 alone (Fig 4E, P < 0.05). Since both integrin $\alpha_{IIb}\beta_3$ activation and P-selectin expression were only incrementally altered by co-incubation of gal-1 and CXCL4, it can be assumed that the elevated platelet aggregation response is caused by additive effects of gal-1 and CXCL4, rather than a synergistic effect induced by possible gal-1–-CXCL4 interactions.

## Galectin-1 does not appear to signal through sialic acid moieties for platelet responses

Gal-1 is known to interact, among others, with polysaccharide moieties on platelets, thereby triggering platelet activation [10], however, the exact mechanism of activation is unknown.

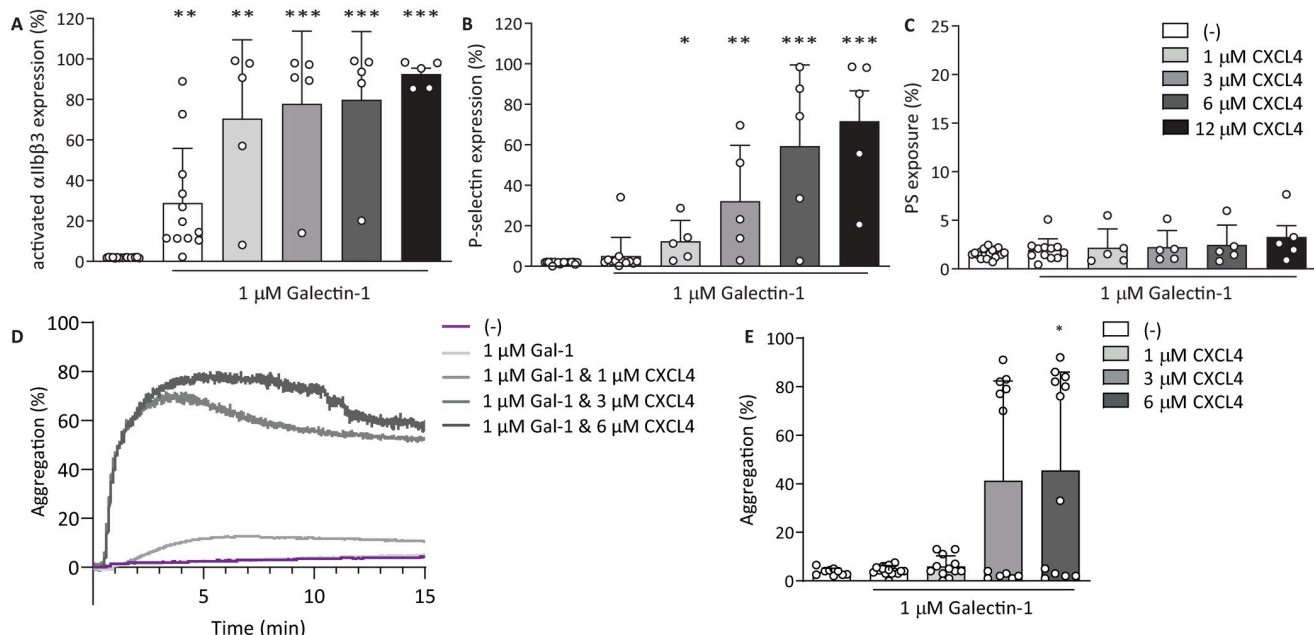

**Fig 4. Co-incubation of galectin-1 and CXCL4 causes more P-selectin expression and faster aggregation.** Gal-1 and CXCL4 were co-incubated on ice for 60 minutes prior to addition to platelets. (A,B,C) Flow cytometry analysis of platelets activated with 1 $\mu$M gal-1 alone or co-incubated with 1, 3, 6, or 12 $\mu$M CXCL4. Integrin $\alpha_{\mathrm{IIb}}\beta_3$ activation (A), P-selectin expression (B) and PS exposure (C) were assessed using fluorescently labelled PAC-1, anti-CD62-P or Annexin A5, respectively. Light transmission aggregometry of washed platelets (D) and quantification (E) incubated with gal-1 and CXCL4. Bars represent mean ± SD (n ≥ 3), *p < 0.05, **p < 0.01, as compared to 1 $\mu$M gal-1 (Kruskal Wallis/Dunn's test).

Therefore, we investigated whether integrin $\alpha_{\mathrm{IIb}}\beta_3$ activation and P-selectin expression were affected after the removal of $\alpha$2-3–$\alpha$2-6-sialic acid side chains by neuraminidase, and if responses to co-stimulation by gal-1 and CXCL4 would be influenced. For this purpose, platelets were treated with neuraminidase and their responses to gal-1 and CXCL4 were measured by flow cytometry. Treatment with neuraminidase led to increased levels of platelet activation markers and lower platelet activation thresholds (Fig 5). In agreement with this, maximum $\alpha_{\mathrm{IIb}}\beta_3$ integrin activation and P-selectin expression were induced at lower levels of gal-1 after neuraminidase treatment (Fig 5Ai and 5Aii). In addition, a higher baseline PS-exposure was induced by sialic acid removal and this PS-exposure was more than doubled after stimulation with gal-1 (Fig 5Aiii). As mentioned above, there was a low level of $\alpha_{\mathrm{IIb}}\beta_3$ integrin activation after neuraminidase treatment, which did show a small increase with CXCL4 (Fig 5Bi). However, even in the presence of 12 $\mu$M CXCL4, this CXCL4-induced increase was not statistically significant compared to neuraminidase-treated platelets alone (Fig 5Bi). Expression of P-selectin and PS-exposure induced by CXCL4 were not increased after treatment with neuraminidase (Fig 5Bii and 5Biii). The activation of integrin $\alpha_{\mathrm{IIb}}\beta_3$ and expression of P-selectin by the combination of gal-1–CXCL4 was not affected by neuraminidase treatment, neither was PS-exposure (Fig 5C). Taken together, these experiments indicate that neuraminidase-treated platelets display pre-activation as evidenced by PS-exposure, which corresponds with previous studies that show an influence of neuraminidases on platelet activity [30, 31]. Further, a lower concentration of gal-1 was sufficient for integrin activation. This indicates that gal-1 primarily induces platelet activation via binding to glycated surface molecules and that this can be modulated by surface sialic acid content. To investigate a possible influence of the altered platelet activation markers on platelet aggregation, LTA was performed with platelets pre-treated with neuraminidase. In agreement with flow cytometry measurements of integrin $\alpha_{\mathrm{IIb}}\beta_3$ activation

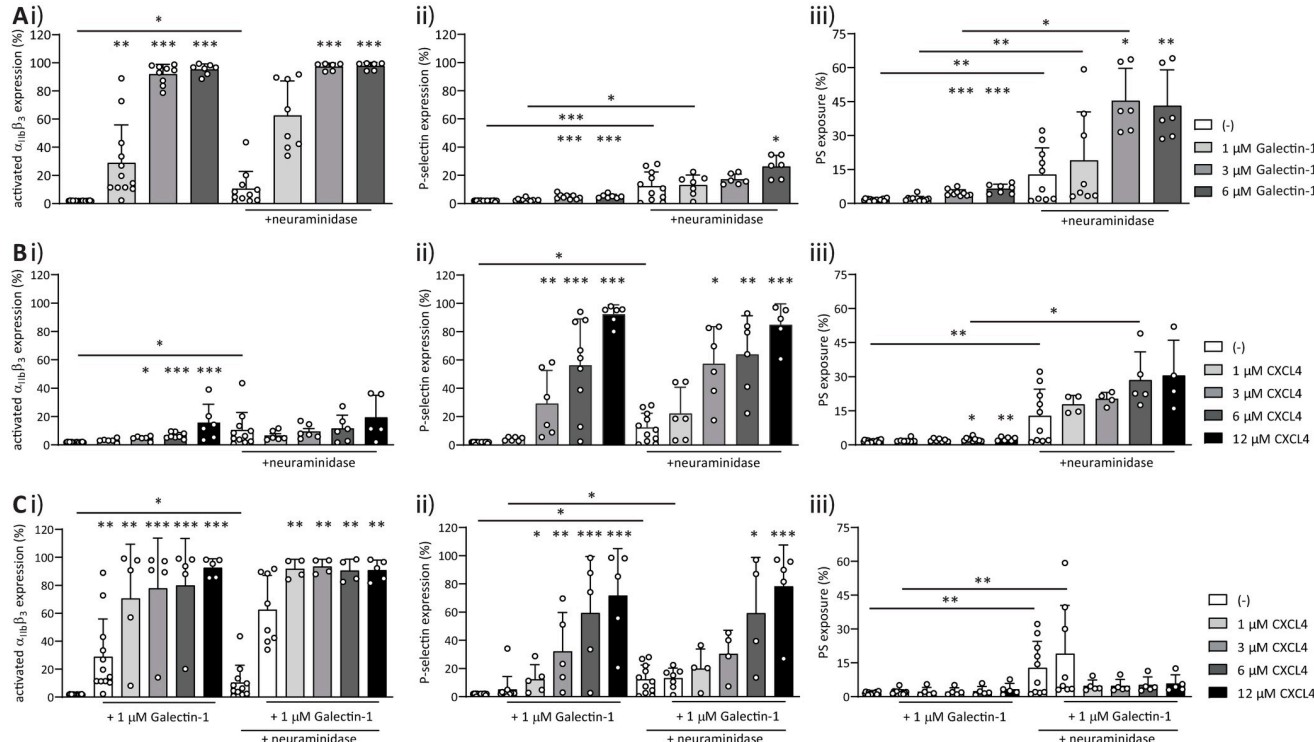

**Fig 5. Neuraminidase-treatment causes more inside-out signaling by galectin-1, unaltered CXCL4 response and a higher PS-exposure independent of either gal-1 or CXCL4.** Platelets were incubated for 60 min at 37˚C without or in the presence of neuraminidase (200 mU per $10^9$ platelets). Flow cytometry measurement of integrin $\alpha_{IIb}\beta_3$ (i), P-selectin (ii) and PS-exposure (iii) of platelets treated with 1, 3, or 6 µM gal-1 (A), 1, 3, 6, or 12 µM CXCL4 (B), or different concentrations of CXCL4 incubated with 3 µM gal-1 (C). Bars represent mean ± SD (n ≥ 3), *p < 0.05, **p < 0.01, ***p < 0.001 as compared to vehicle within appropriate treatment with or without neuraminidase or conditions compared between treatments (Kruskal Wallis/Dunn's test).

and P-selectin expression, robust platelet aggregation was induced by lower concentrations of gal-1 after neuraminidase treatment (Fig 6A). Interestingly, induction of platelet aggregation by CXCL4 was completely abolished after neuraminidase treatment (Fig 6B), indicating CXCL4 may pass signals through sialic acid-modified surface proteins. However, P-selectin expression and $\alpha_{IIb}\beta_3$ integrin activation were unaltered following CXCL4 stimulation after neuraminidase treatment (Fig 6B). Aggregation after co-stimulation with gal-1 and CXCL4 was unaltered (Fig 6C) and likely caused by the addition of gal-1 at 3 µM, which alone already resulted in maximal aggregation.

## Galectin-1 binds to normal and desialated platelets

Considering the increased reactivity towards gal-1 after removal of sialic acid, it was examined whether surface sialic acid content modulates gal-1 binding. Thus, the binding of fluorescently-labelled gal-1 titrated to platelets was monitored using flow cytometry. First, the binding of gal-1 to washed platelets pre-treated with neuraminidase or vehicle was compared. In accordance to the increased integrin activation and P-selectin expression and increased aggregation (cf. Figs 5 and 6), there was increased binding of gal-1 to desialylated platelets (Fig 7A). Next the influence of CXCL4 on the binding of gal-1 to platelets was assessed. Pre-incubation of CXCL4 and gal-1 together before addition to platelets did not alter the binding to platelets, compared to gal-1 alone (Fig 7B). Interestingly, the increased binding of gal-1 to desialylated platelets appeared to be diminished in the presence of CXCL4, which suggests that CXCL4

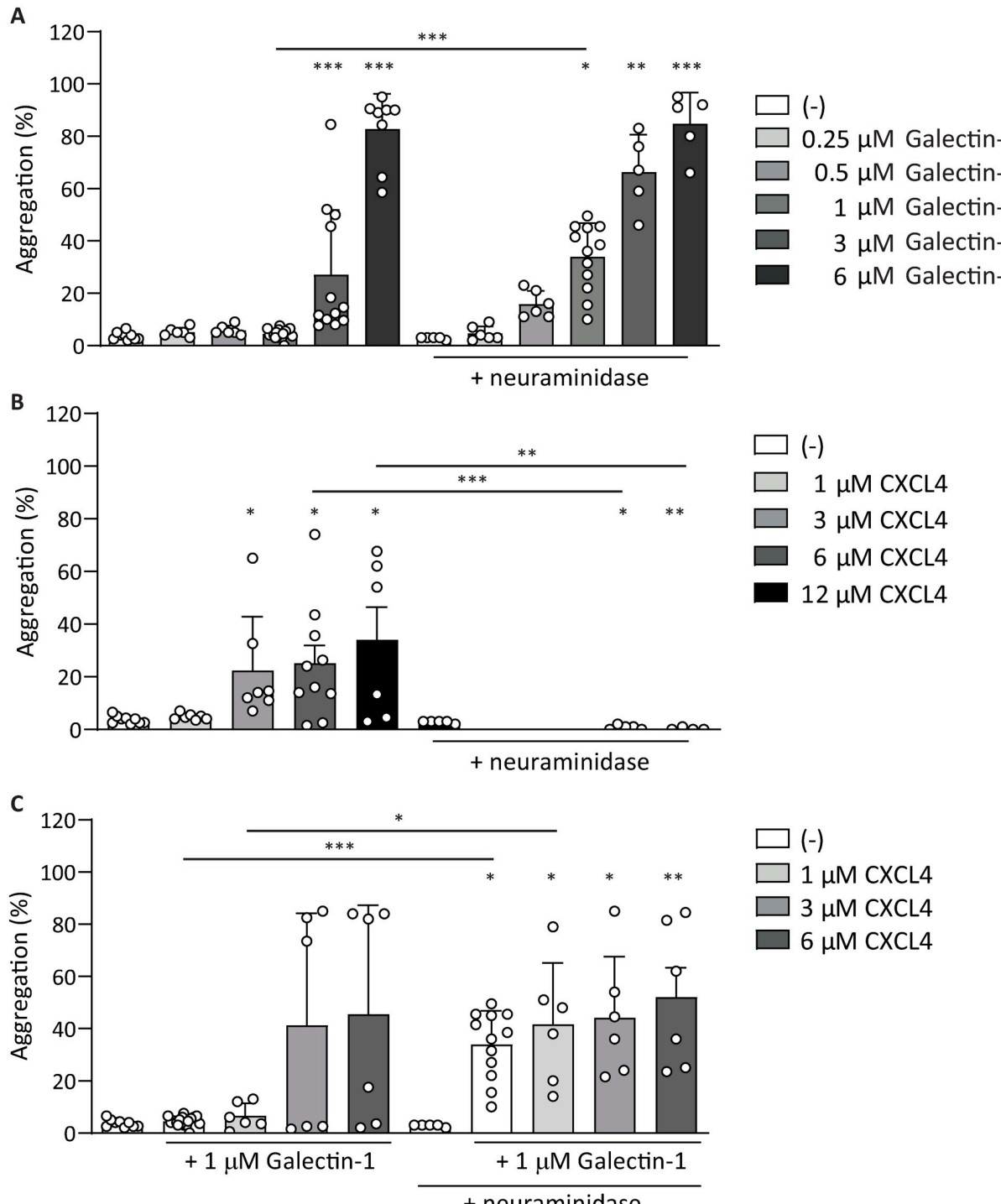

**Fig 6. Neuraminidase-treatment causes platelet aggregation at lower galectin-1 concentrations.** Platelet aggregation was induced in washed platelets by different concentrations of gal-1 (A), CXCL4 (B), or different concentrations of CXCL4 co-incubated with 1 μM gal-1 (C), with or without pretreatment with neuraminidase. Bars represent mean ± SD (n ≥ 3), $^*p < 0.05$, $^{**}p < 0.01$ as compared to vehicle within appropriate treatment with or without neuraminidase or conditions compared between treatments (Kruskal Wallis/Dunn's test).

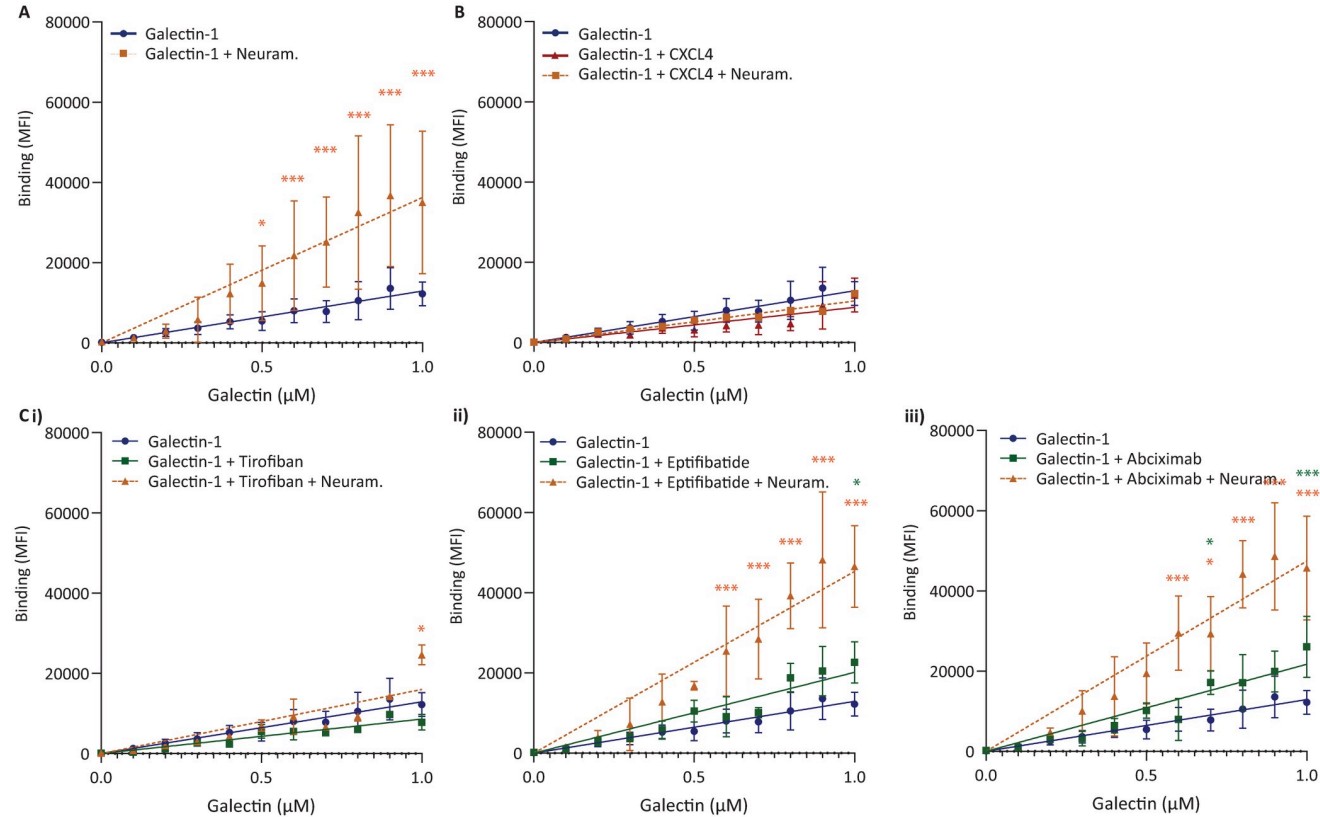

**Fig 7. Galectin-1 binds to platelets independent of sialic acid or integrin $\alpha_{IIb}\beta_3$.** Washed platelets were incubated for 60 min at 37˚C, with or without the presence of neuraminidase (200 mU/$10^9$ platelets), and next with or without the integrin $\alpha_{IIb}\beta_3$-inhibitor tirofiban (1 µg/mL). (A) Flow cytometry analysis of OG-labelled gal-1 bound to platelets with (yellow ■) or without neuraminidase (blue •). (B) Analysis of gal-1 (blue •) compared to gal-1+CXCL4 (red ▲) or gal-1+CXCL4 on neuraminidase-pretreated platelets (orange ■). Analysis of gal-1 (blue •) bound to platelets in the presence of (C) tirofiban (green ■) or tirofiban on neuraminidase-pretreated platelets (orange ▲), (D) Eptifibatide (green ■) or eptifibatide on neuraminidase-pretreated platelets (orange ▲), (E) Abciximab (green ■) or abciximab on neuraminidase-pretreated platelets (orange ▲). Dots represent mean ± SEM, n = 3. *p < 0.05, **p < 0.01, ***p < 0.001 as compared to 3 µM Galectin-1 (Kruskal Wallis/Dunn's test).

might occupy potential binding sites for gal-1 on desialylated platelets. As integrin $\alpha_{IIb}\beta_3$ has been described as a putative receptor for gal-1 [9], we compared the binding of gal-1 on platelets that have been pretreated with tirofiban, eptifibatide and abciximab, $\alpha_{IIb}\beta_3$ integrin inhibitors, after neuraminidase treatment (Fig 7C–7E). Interestingly, tirofiban diminished the binding of gal-1 only to desialylated platelets (Fig 7A and 7C). Eptifibatide and abciximab were found to not alter the binding of gal-1 to both untreated and neuraminidase-treated platelets, indicating either that gal-1 does not bind to integrin $\alpha_{IIb}\beta_3$ or, more plausible, does not share a binding site with these inhibitors. In order to investigate whether gal-1 could influence the binding of CXCL4 to heparin, CXCL4 was perfused in increasing concentrations over immobilized heparin using SPR. Whereas CXCL4 bound avidly to heparin, this binding was not influenced by the presence of a >10-fold molar excess of gal-1 (S3A and S3B Fig). To investigate whether gal-1 could influence the binding of the HIT model antibody KKO to CXCL4 in complex with heparin, KKO was perfused in increasing concentrations over the immobilized CXCL4/heparin complex in the absence or presence of gal-1 (S3C–S3F Fig). The presence of gal-1 did not influence the binding of KKO to CXCL4/heparin, suggesting that gal-1 does not affect the formation of HIT immune complexes (S3D–S3F Fig).

## Discussion

The importance of protein-glycan interactions in homeostasis, neoplasia and immunity is becoming increasingly clear [32–34]. Galectins serve as integral mediators of these protein-glycan interactions [32, 34, 35]. Interestingly, in a recent study it was found that gal-3 on tumor cells interacts with glycoprotein VI on platelets, thereby promoting tumor metastasis [36]. Also in thrombosis and hemostasis, galectins have a modulatory role. For example, a study using mouse models showed that gal-3 expression was induced during early venous thrombosis and the protein presented on the venous wall, red blood cells, platelets and microvesicles [37]. Circulating levels of gal-3 correlated with thrombus size and genetic deletion of gal-3 reduced thrombosis, which could in turn be rescued by exogenous gal-3 treatment [37]. In addition, gal-3 has recently been found to bind to the chemokine CXCL12, resulting in an inhibition of CXCL12's chemotactic functions [38]. Interestingly, both gal-1 and gal-3 serve as binding partners for circulating coagulation factor VIII and von Willebrand Factor [12, 13]. Although the binding of gal-1 and -3 appeared to downregulate the function of these coagulation factors, these galectins appear to promote thrombosis and hemostasis, as genetic knockdown reduces clot size [37] and prolongs bleeding time [9]. Besides gal-1 and gal-3, gal-8 is also expressed by endothelial cells and binds platelets through glycans, triggering platelet aggregation, spreading and granule secretion [39]. As well as being a platelet activator, platelets release intracellular gal-8 when triggered by thrombin. This study confirms previous observations that gal-1 serves as a physiologic agonist for platelets [9]. However, although activation of platelets by gal-1 led to an increased release of CXCL4, which co-localized with gal-1 on the platelet surface, endogenous CXCL4 appeared to be unable to activate platelets by itself (see below). In addition, serotonin release from dense granules was not induced by gal-1. Interestingly, whereas treatment of platelets with exogenous gal-1 led to robust expression of the active conformation of integrin $\alpha_{IIb}\beta_3$ and platelet aggregation, but not to elevated expression of CD62P, addition of CXCL4 did not result in notable integrin $\alpha_{IIb}\beta_3$ activation nor in noteworthy platelet aggregation, but to a pronounced expression of P-selectin. This suggests that gal-1 and CXCL4 have opposite, yet complementary activities, and would act in concert to achieve full platelet activation, i.e. aggregation, granule release and integrin activation. However, the exposure of PS, and thus the formation of a pro-coagulant platelet response, might require additional stimuli. The observation in this study that gal-1 did not induce P-selectin expression is different from previous observations [9, 10]. The exact reason for this discrepancy is not known, but might be explained by donor-to-donor variations and different duration of activation. In addition, it is surprising that gal-1 induced the release of CXCL4, but not of P-selectin in our study. A possible explanation could be that CXCL4 and P-selectin are packed in different $\alpha$-granules, which are differentially released upon stimulation with gal-1 or CXCL4, alike the angiogenic regulators endostatin and VEGF [40], and the proteins VWF and fibrinogen as shown previously [41]. This might also explain why gal-1 induced platelet aggregation more effectively than CXCL4. Blockade of possible endogenously released CXCL4, induced by gal-1, by heparin, did not influence the expression of platelet activation markers, suggesting that CXCL4 release from platelets does not influence the course of gal-1–induced platelet activation. Besides being a more robust activator of an active conformation of integrin $\alpha_{IIb}\beta_3$, gal-1 might also trigger a more effective release of fibrinogen from the washed platelets. Unlike collagen, gal-1 did not induce platelet aggregation in plasma up to 6 $\mu$M. Plasma contains several abundant proteins that might sequester gal-1 (IgM, $\alpha$2-macroglobulin, fetuin) [42]. In addition, plasma may also contain unspecified interacting agents that compete with the carbohydrate receptors for gal-1 on platelets. Previous studies have reported serum and plasma concentrations of gal-1 in healthy individuals ranging from 1-17 nM [43, 44]. However, given

the potential interaction of gal-1 with several cellular components of the vasculature, the measured soluble levels of gal-1 might be an underestimation of the total gal-1 present in a blood vessel (similar to the anticoagulant TFPI, the majority of the vascular pool is bound to the vessel wall [45]). Concentrations of up to 2.4 $\mu$M have been observed within the bovine spleen [46], indicating that local concentrations might reach orders of magnitude above those in plasma and might even increase further during inflammation. The concentrations used in the experiments in this study are in the range of those used in previous studies [9, 10] and is around the physiologic monomer-dimer equilibrium constant of gal-1 (7$\mu$M, [27]). Nevertheless, despite the relatively high concentrations investigated and the lack of response of platelets to gal-1 in a plasma environment, studies in gal-1-deficient mice showed altered platelet responses e.g. in a model of bleeding time. This suggests that gal-1 can reach sufficiently high concentrations and can be active in a plasma environment. Since gal-1 and CXCL4 are present in and released from platelets, they might act in concert at sites of vascular injury.

CXCL4 has various roles in the regulation of hemostasis [17, 18, 47, 48]. Most of these activities can be explained by its high positive charge and the exceptionally high affinity of CXCL4 for negatively charged biomolecules [49]. A single cognate receptor for CXCL4 has been elusive thus far, although CXCL4 has been proposed to bind to several putative receptors [50–53]. Although platelets have been known to express functional chemokine receptors [54, 55], the possibility of platelet activation through these receptors by CXCL4 has not been extensively studied in literature. A previous study by Kowalska and colleagues has reported no platelet aggregation after stimulation with CXCL4 at undisclosed concentrations [54]. This is in line with the observations in this study, as platelet aggregation was only observed at the highest concentration of CXCL4 and only after 15 minutes. The poor aggregation in response to CXCL4 is also paralleled by the virtual absence of $\alpha_{IIb}\beta_3$ activation. However, CXCL4 was shown to induce robust surface P-selectin expression at micromolar amounts, which is in the range of ADP-, TRAP6- and epinephrin-induced platelet activation [56]. Being mainly P-selectin externalization, CXCL4 appears to induce a rather atypical platelet response. The question concerning the receptor for CXCL4 on platelets remains to be addressed. Previous studies have excluded expression of CXCR3 on platelets [55] and although CCR1 is expressed on platelets in low amounts, stimulation of CCR1 by CCL5 rather induced platelet aggregation [55]. Thus, the observations in this study might be explained by general electrostatic interactions of CXCL4 with negatively charged polyanions present on the platelet surface. Interactions with polyanions are also crucial for the pathophysiology of HIT. The SPR results indicate that neither the interaction of CXCL4 with heparin, nor the binding of the model antibody KKO to CXCL4/heparin is influenced by gal-1. Thus, it is unlikely that gal-1 contributes to the development of HIT.

Sialic acid expression on the platelet surface has been shown to play an important role in platelet lifespan and clearance from circulation [57]. Desialylation of platelets leads to rapid clearance by the liver through the action of $\beta$2-integrins on macrophages and the hepatic Ashwell-Morell receptor [58]. The extent of sialic acid presentation on the platelet surface has been shown to have implications for platelet functions, although it appears somewhat controversial whether platelet activity increases [59] or decreases [30]. Also endogenous neuraminidases have recently been shown to modulate platelet activation [31]. In the current study, reactivity of platelets towards gal-1 was increased after enzymatic removal of surface $\alpha$2-3- and $\alpha$2-6-sialic acid moieties. The concentration of gal-1 needed to induce a platelet response was decreased approximately 2-3 fold. Interestingly, while neither gal-1 nor CXCL4 induced notable PS exposure in untreated platelets, it was potently induced by either protein after neuraminidase treatment. Baseline levels of PS-exposure where also increased, indicating that sialic acid removal by itself might induce platelet activation, or that desialylated

platelets have a lower activation threshold. Previous studies have shown that sialic acid removal strongly increased the subsequent induction of PS-exposure (apoptosis) by gal-1 in T-cells and HL-60 myeloid leukemia cells [60, 61]. This observation was explained by the "unmasking" of gal-1-binding sugar moieties on the cell surface, thereby increasing the number of surface binding sites and allowing potentiation of cell activation. Similar mechanisms might underlie the observations in this study, since removal of sialic acid led to an increased binding of fluorescently labelled gal-1 to the platelet surface. Binding of lectins (e.g. Ashwell-Morell receptor) to platelets plays an integral role in platelet clearance and it may be speculated that galectins play a role in this process as well. However, desialylation of platelets also resulted in a complete abrogation of platelet aggregation after addition of CXCL4. Since it is yet unknown what molecules actually mediate binding of CXCL4 to platelets, one can only speculate about the mechanistic background of this observation. Since CXCL4 is a weaker platelet stimulant than e.g. collagen, and glycosaminoglycans mediate binding to CXCL4, it might be an unspecific effect that is a result of glycan modification by neuraminidase. Although integrin $\alpha_{IIb}\beta_3$ has been found to play a role in the binding of gal-1 to platelets, the addition of $\alpha_{IIb}\beta_3$ antagonists did not reduce gal-1 binding, except for tirofiban to desialylated platelets. Being small molecule antagonists that specifically target the $\alpha_{IIb}\beta_3$ RGD-binding site, tirofiban and eptifibatide might not effectively compete with gal-1, as gal-1 might have a different (glycan-based) binding site on integrin $\alpha_{IIb}\beta_3$. Abciximab might also bind to a site that is different from gal-1. Romaniuk et al. have shown that gal-1 binding to platelets and gal-1-induced platelet activation is reduced, but not abolished, by CD41 antibodies, and on platelets from patients with Glanzmann thrombasthenia, meaning that integrin $\alpha_{IIb}\beta_3$ might not be the only receptor for gal-1 on platelets [9]. Although a single surface receptor for gal-1 has not been specified and gal-1 binds to a number of glycated ligands, it appears that gal-1 binding to cells generally leads to the exposure of PS. Possible receptors other than $\alpha_{IIb}\beta_3$ might be platelet glycoproteins GPIb$\alpha$ and GPVI, due to their glycosylation. However, the findings in this study with the GPVI agonists collagen and CRP-XL indicate that the pattern of platelet activation is different, which would speak against GPVI as a bona-fide receptor for gal-1. In addition, unlike gal-3 which was found to bind to GPIV, gal-1 does not have a collagen-like domain on which the interaction with GPVI is likely based [36]. Based on the heavy glycosylation of GPIb$\alpha$, the possibility of flow-dependent binding of platelets to immobilized gal-1 is conceivable, yet future studies are necessary to establish this. Co-addition of CXCL4 did not enhance gal-1 binding to platelets and even reduced binding of gal-1 to desialylated platelets. Given the preference of CXCL4 and gal-1 for sulfated glycans [21, 61], it is conceivable that CXCL4 and gal-1 at least in part share negatively charged binding sites on the platelet surface. Alternatively, Gal-1 and CXCL4 might undergo heterophilic interactions, as was recently shown for Gal-3 and CXCL12 [38], leading to an altered preference for particular glycans.

## Conclusion

CXCL4 and gal-1 are demonstrated to induce a complementary platelet activation, with CXCL4-stimulation leading to the expression of P-selectin and gal-1-stimulation to platelet aggregation. While the induction of PS-exposure and thus procoagulant function of platelets after treatment with either CXCL4 and gal-1 was minor, enzymatic desialylation resulted in potent induction of procoagulant surface after stimulation with CXCL4 and gal-1. Given their presence in platelets and their emerging role of platelets and their contents in inflammation [62, 63], CXCL4 and gal-1 may act in concert to modulate platelet-mediated host defence and immune processes.

## Supporting information

**S1 Fig. Galectin-1–induced platelet responses in the presence of heparin.** Bar graphs represent the percentage of platelet $\alpha_{IIb}\beta_3$ activation (A), P-selectin expression (B) and PS-exposure (C) by gal-1 in the absence or presence of heparin ($10\mu g/mL$). Mean ± SD (n = 4-12). $^*p < 0.05$, $^{**}p < 0.01$, $^{***}p < 0.001$ as compared to control (no gal-1, Kruskal Wallis/Dunn's test).
(DOCX)

**S2 Fig. Platelet aggregation in platelet-rich plasma.** Platelet aggregation was induced in PRP by increasing concentrations of gal-1 or 5 $\mu g/mL$ collagen. Bars represent mean ± SD (n = 3).
(DOCX)

**S3 Fig. Surface plasmon resonance of CXCL4 and antibody KKO on heparin.** SPR analysis of CXCL4 binding on immobilized unfractionated heparin in the absence (A) or presence (B) of 500 nM gal-1. C: Sensorgram of the experimental course of KKO binding to CXCL4/heparin. CXCL4 was immobilized (i), obtaining a stable baseline (x), then KKO was perfused (ii) followed by a dissociation phase (iii). The arrows denote start and end of KKO perfusion and start of the wash phase with perfused heparin. Inset: Binding of KKO to heparin alone. D: Binding response (in resonance units, RU) of increasing concentrations of KKO (0-125 nM) in the absence (black dots) or presence (red squares) of gal-1 (500 nM). Representative sensorgrams of KKO binding to CXCL4/heparin in the absence (E) or presence (F) of gal-1.
(DOCX)

## Acknowledgments

The authors thank Alexandra Heinzmann for expert help with the serotonin ELISA.

## Author Contributions

**Conceptualization:** Johan W. M. Heemskerk, Victor L. J. L. Thijssen, Marijke J. E. Kuijpers, Rory R. Koenen.

**Data curation:** Annemiek Dickhout, Bibian M. E. Tullemans.

**Formal analysis:** Annemiek Dickhout, Bibian M. E. Tullemans, Marijke J. E. Kuijpers.

**Funding acquisition:** Johan W. M. Heemskerk, Victor L. J. L. Thijssen, Rory R. Koenen.

**Investigation:** Victor L. J. L. Thijssen, Marijke J. E. Kuijpers.

**Resources:** Johan W. M. Heemskerk, Marijke J. E. Kuijpers, Rory R. Koenen.

**Supervision:** Victor L. J. L. Thijssen, Marijke J. E. Kuijpers, Rory R. Koenen.

**Writing – original draft:** Annemiek Dickhout, Victor L. J. L. Thijssen, Rory R. Koenen.

**Writing – review & editing:** Bibian M. E. Tullemans, Johan W. M. Heemskerk, Marijke J. E. Kuijpers.

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
