## [Decision Letter · Decision Letter 0]

29 Jun 2020

PONE-D-20-16580

Galectin-1 and platelet factor 4 (CXCL4) induce complementary platelet responses.

PLOS ONE

Dear Dr. Koenen,

Thank you for submitting your manuscript to PLOS ONE. After careful consideration, we feel that it has merit but does not fully meet PLOS ONE’s publication criteria as it currently stands. Therefore, we invite you to submit a revised version of the manuscript that addresses the points raised during the review process.

Specifically, please compare platelet responses to galectin-1 in platelet rich plasma (PRP) and whole blood. A common platelet agonist should be added as positive control and reference. Please address experimentally, or at least discuss, potential discrepancies in the release of different secretory proteins (refers to P-selectin). Please discuss in more depth potential translational aspects of this study, including plasma levels.

We look forward to receiving your revised manuscript.

Kind regards,

Christian Schulz

Academic Editor

PLOS ONE

Journal Requirements:

3.Thank you for stating the following in the Acknowledgments Section of your manuscript:

[This work was supported by the Netherlands Foundation for Scientific Research

(ZonMW VIDI 016.126.358), the Landsteiner Foundation for Blood Transfusion

Research (LSBR Nr. 1638) awarded to Rory R. Koenen. Support from CARIM to

Annemiek Dickhout is highly acknowledged. Marijke J.E. Kuijpers was supported by

the Maastricht Thrombosis Expertise Centre as part of the Heart+Vascular Centre

(HVC) of the MUMC+.]

 [Netherlands Foundation for Scientific Research (ZonMW VIDI 016.126.358), the Landsteiner Foundation for Blood Transfusion Research (LSBR Nr. 1638) awarded to R.R.K.

www.zonmw.nl, www.lsbr.nl

The funders had no role in study design, data collection and analysis, decision to publish, or preparation of the manuscript.]

Reviewers' comments:

Reviewer's Responses to Questions

**Comments to the Author**

1. Is the manuscript technically sound, and do the data support the conclusions?

Reviewer #1: Yes

Reviewer #2: Partly

2. Has the statistical analysis been performed appropriately and rigorously? 

Reviewer #1: Yes

Reviewer #2: Yes

3. Have the authors made all data underlying the findings in their manuscript fully available?

Reviewer #1: Yes

Reviewer #2: Yes

4. Is the manuscript presented in an intelligible fashion and written in standard English?

Reviewer #1: Yes

Reviewer #2: Yes

5. Review Comments to the Author

Reviewer #1: This is an interesting experimental study providing a further insight into the molecular mode of action of galectin-1 and CXCL4 in platelet activation aiming the questions whether they act in synergy or in a complementary manner. The authors apply a sufficient spectrum of experimental approaches to follow this concept and provide a convincing introdution for the readers into that topic. Although the study is well performed and data presentation and interpretation is mainly clear, some minor concerns remain to be addressed:

Concerning the experimental strategy:

- The authors show that gal-1 induced the release of CXCL4. Although the amounts of CXCL4 released (Figure 1) appear to be below the activity level shown in Figure 3, an isolated consideration of gal-1 and CXCL4 activities could be achieved by blocking released CXCL4 using a decoy approach. Have the authors tried that? In light of the high concentrations of CXCL4 mentioned (line 36-37) that could be released, this would be a valuable appraoch, that, if not performed, at least could be mentioned in the discussion.

- Why didn't you check other, easy to perform platelet activation readouts, such as alpha granule release (ATP), to further differentiate the activities of the two components? Have you tried this?

Concerning the experimental details:

- The integrin activation studies (Figure 2A, 3A) have been performed by binding the PAC-1 antibody, but data are given in 2E and 3E as expression in %. How is this calculated and what is 100% control?

- Why do the authors stop in Fig. 3D the aggregation curve just in the on-set of aggregation? Why not taking a later time point than 15 min for comparing the aggregation outcome?

- Figure 4E, the individual data display a strong deviation and it appears questionable to derive a statistics from this data set. How do these data correspond with the curve in Figure 4D, is this a representative one?

-When blocking the GPIIbIIIa to check a potential receptor function for gal-1, have the autors tried eptifibatide for that, which appears more promising in light of the greater expansion .

- please add statistics in the legend of Figure 6

Other minor points to be addressed:

- GPIb, could that be a probable receptor for gal-.1 considering the high glycosylation of GPIb, has this been postulated in literature and is this worth to considerer as option to be mentioned in discussion

- line 35; 2 % of mass; mass of platelets or mass of granule content?

- Mat Meth section, the authors inconsistently mentioned the sources, company, partly the location and country, please make it more consistent

- minor misprints line 13, 280,

Reviewer #2: In this study Dickhout et al. investigate synergistic effects of galectin-1 and CXCL4 on platelet activation and aggregation using an in vitro approach. The authors found that exogenously added galectin-1 and CXCL4 exerted additive effects on platelet aggregation with galectin-1 leading to integrin activation and CXCL4 mediating P-Selectin surface expression. Desialysation by neuraminidase lowered thresholds for galactin-1- and CXCL4-mediated effects and additionally induced a pro-coagulatory platelet phenotype. Further, neuraminidase treatment abolished CXCL4-induced pro-aggregatory effects, while CXCL4 prevented increased galectin-1 binding induced by neuraminidase.

This is a small in vitro study, experiments are straightforwardly designed and technically mostly well performed. The hypothesis is interesting in principle as galectins and CXCL4 both are interesting players in thrombo-inflammation. However, due to the in vitro-only design and exclusively exogenously added galectin and CXCL4, pathophysiological relevance of the findings remains somewhat speculative. Despite the lack of a straightforward translational aspect, for people in the field this study would still add something to the current state of knowledge. Accordingly, I would support publication in this journal after a major revision of the article, if the authors adequately address my comments/questions listed below.

Major comments/questions:

1. The authors should discuss more concretely for what pathophysiological conditions their findings might be relevant. Could galectin-1 possibly trigger/propagate heparin-induced-thrombocytopenia (HIT) where CXCL4 plays a key pathophysiological role?

2. Does galectin-1 induce similar platelet responses in platelet rich plasma (PRP) and whole blood?

3. Can concentrations of gal-1 that induced platelet activation in vitro be reached in human blood under some circumstances?

4. In the discussion interaction of gal-3 with GPVI on platelets is mentioned. Could a similar mechanism be responsible for galectin-1 mediated effects? Authors should check how blocking of GPVI or co-stimulation with GPVI-ligands (collagen or CRP) affect galectin-1 mediated platelet activation and aggregation.

5. Galectin-1 induced CXCL4 release and surface expression, however (according to results shown in figure 2) this was not able to induce the effects exerted by exogenously added CXCL4. Does endogenous CXCL4 play a role for galectin-1 mediated effects? Given the fact, that platelet-derived CXCL4 would not be sufficient for the observed additive effects with galectin-1, what would be the main source of such concentrations of exogenous CXCL4 in vivo?

6. In figure 2 and 3 a positive control (e.g. a classical platelet activator) should be included for each experiment. This is particularly important for P-Selectin in figure 2, as not only conflicting results to previously reported data have been observed, but results stay also in some conflict with the observations that galectin-1 induced CXCL4-release which is also supposed to be stored in alpha granules. The authors might consider to validate the results using a second P-Selectin antibody.

7. To investigate gal-1 binding to platelet integrin-receptor after neuraminidase treatment authors should make additional experiments using antibody-based integrin-inhibitors (e.g. Abciximab) rather than only small molecule inhibitor Tirofiban.

8. Authors should discuss in more detail why neuraminidase abolishes CXCL4-induced platelet aggregation but on the other side lowered threshold for P-Selectin expression.

Minor comments:

1. In figure 1 one could assume some co-localization of galectin-1 and CXCL4, however the cross-section used for intensity profile does not seem representative for the figure. Increased image quality would strengthen the results, additionally for the PF4-antibody a respective isotype control image should be demonstrated to show antibody specificity.

2. The experimental design in figure 4 is not appropriate to investigate influence of CXCL4 on galectin-1-induced integrin activation. Further effects of increasing concentrations of CXCL4 cannot be excluded as 3 µM of galectin-1 already showed almost maximal activation. Experiment should be repeated using a lower dose of galectin-1 or the conclusion and interpretation must be revised.

3. Result section related to figure 4 (page 8, line 150-160) is confusing and somewhat contradictory.

4. It is not always clear whether galectin-1, CXCL4 and neuraminidase were added simultaneously or which treatment was first.

5. Y-axis in graphs showing PAC-1 expression should be labeled “activated integrin expression”.

6. PLOS authors have the option to publish the peer review history of their article (what does this mean?). If published, this will include your full peer review and any attached files.

Reviewer #1: No

Reviewer #2: No

---

## [Author Response · Author response to Decision Letter 0]

8 Dec 2020

Point-by-point response

Reviewer #1: This is an interesting experimental study providing a further insight into the molecular mode of action of galectin-1 and CXCL4 in platelet activation aiming the questions whether they act in synergy or in a complementary manner. The authors apply a sufficient spectrum of experimental approaches to follow this concept and provide a convincing introduction for the readers into that topic. Although the study is well performed and data presentation and interpretation is mainly clear, some minor concerns remain to be addressed:

We wish to thank the reviewer for the accurate review and for the constructive comments on our work. By addressing the points raised below we hope to have sufficiently improved our study.

Concerning the experimental strategy:

- The authors show that gal-1 induced the release of CXCL4. Although the amounts of CXCL4 released (Figure 1) appear to be below the activity level shown in Figure 3, an isolated consideration of gal-1 and CXCL4 activities could be achieved by blocking released CXCL4 using a decoy approach. Have the authors tried that? In light of the high concentrations of CXCL4 mentioned (line 36-37) that could be released, this would be a valuable approach, that, if not performed, at least could be mentioned in the discussion.

We thank the reviewer for raising this point. We were also aware of this and we have considered neutralizing CXCL4 in the presence of gal-1. From our own experience, CXCL4 is difficult to neutralize using antibodies. In fact, this might even result in the formation of immune complexes that activate platelets (Nguyen et al., doi: 10.1038/ncomms14945). Thus, we have chosen heparin as a CXCL4 neutralizing agent, since several studies including our own have shown that heparin is able to block CXCL4 function (Vajen et al., doi: 10.1080/20013078.2017.1322454.). We now show results of experiments in which we added heparin to our experimental setup. The presence of heparin appeared to somewhat sensitize the platelets, yet there were no differences in the course of platelet activation by gal-1. We have now included these data as S1_Fig in the revised manuscript and these were discussed on page 8, line 179.

- Why didn't you check other, easy to perform platelet activation readouts, such as alpha granule release (ATP), to further differentiate the activities of the two components? Have you tried this?

Thank you for this valid point. We have now included the requested data in Figure 1B and we have discussed the findings on page 7 line 157. 

Concerning the experimental details:

- The integrin activation studies (Figure 2A, 3A) have been performed by binding the PAC-1 antibody, but data are given in 2E and 3E as expression in %. How is this calculated and what is 100% control?

We have indeed omitted to clarify this in the methods section. The data was expressed as percentage fluorescence-positive platelets of the total platelet count, with the marker set at approx. 0.67 times the fluorescence of quiescent platelets. This was added now on page 5 line 105. 

- Why do the authors stop in Fig. 3D the aggregation curve just in the on-set of aggregation? Why not taking a later time point than 15 min for comparing the aggregation outcome?

In fact, we did measure for up to 24 minutes, but the signals did not change any further after 15 minutes. In order to keep the light transmission data comparable with the flow cytometry data (these were until 15 minutes), we have decided to not show any light transmission data after 24 minutes. We hope that we have clarified this point and that this is acceptable for the reviewer.

- Figure 4E, the individual data display a strong deviation and it appears questionable to derive a statistics from this data set. How do these data correspond with the curve in Figure 4D, is this a representative one?

This is indeed a sharp observation by the reviewer. We explain the observations by a donor-to-donor difference and by our observation that gal-1 is not such a potent trigger as collagen or thrombin. The platelets tend to respond in an all-or-none fashion, at least in aggregation experiments. Following also the comments of reviewer #2, we have replaced the original 3µM gal-1 data in Figure 4 with 1µM gal-1. However similar to our observations with 3µM gal-1, the bimodal responses with 1µM gal-1 combined with 3 and 6 µM CXCL4 remain. We have briefly discussed this observation on page 10 line 210.

-When blocking the GPIIbIIIa to check a potential receptor function for gal-1, have the authors tried eptifibatide for that, which appears more promising in light of the greater expansion.

This valid point has also been raised by reviewer #2. Following this suggestion, we have expanded figure 7 with experimental data implementing both eptifibatide and Reopro. We have discussed the observations on page 12 line 270 and page 17, line 408.

- please add statistics in the legend of Figure 6

We had already described all statistical parameters in the figure legends. We now have also added the method of statistical testing in all legends.

Other minor points to be addressed:

- GPIb, could that be a probable receptor for gal-.1 considering the high glycosylation of GPIb, has this been postulated in literature and is this worth to considerer as option to be mentioned in discussion. 

This is indeed an interesting possibility and we have discussed this now, along with a possible binding of gal-1 to GPVI starting on page 17, line 420.

- line 35; 2 % of mass; mass of platelets or mass of granule content?

We have specified this as 2% platelet protein mass.

- Mat Meth section, the authors inconsistently mentioned the sources, company, partly the location and country, please make it more consistent.

This has been adjusted accordingly.

- minor misprints line 13, 280,Fuctions , at the

Thank you. This has been corrected.

 

Reviewer #2: In this study Dickhout et al. investigate synergistic effects of galectin-1 and CXCL4 on platelet activation and aggregation using an in vitro approach. The authors found that exogenously added galectin-1 and CXCL4 exerted additive effects on platelet aggregation with galectin-1 leading to integrin activation and CXCL4 mediating P-Selectin surface expression. Desialysation by neuraminidase lowered thresholds for galactin-1- and CXCL4-mediated effects and additionally induced a pro-coagulatory platelet phenotype. Further, neuraminidase treatment abolished CXCL4-induced pro-aggregatory effects, while CXCL4 prevented increased galectin-1 binding induced by neuraminidase.

This is a small in vitro study, experiments are straightforwardly designed and technically mostly well performed. The hypothesis is interesting in principle as galectins and CXCL4 both are interesting players in thrombo-inflammation. However, due to the in vitro-only design and exclusively exogenously added galectin and CXCL4, pathophysiological relevance of the findings remains somewhat speculative. Despite the lack of a straightforward translational aspect, for people in the field this study would still add something to the current state of knowledge. Accordingly, I would support publication in this journal after a major revision of the article, if the authors adequately address my comments/questions listed below.

We wish to thank the reviewer for the dedicated review and for the useful suggestions, which we have addressed below. We hope that the reviewer agrees that our study has been improved and that it is now considered acceptable for publication.

Major comments/questions:

1. The authors should discuss more concretely for what pathophysiological conditions their findings might be relevant. Could galectin-1 possibly trigger/propagate heparin-induced-thrombocytopenia (HIT) where CXCL4 plays a key pathophysiological role?

We have performed surface plasmon resonance (SPR) in order to address this valid and interesting comment. First, we addressed the question whether galectin-1 could influence the binding of CXCL4 to heparin. For this, we immobilized heparin on an SPR flow cell and perfused CXCL4 in increasing concentrations, in the absence or presence of a >10fold surplus of galectin-1. No difference in interaction was found. Next, we immobilized CXCL4 on this heparin surface, thereby creating a stable CXCL4/heparin complex baseline. We then compared binding of the HIT model monoclonal antibody KKO (Nguyen et al., doi: 10.1038/ncomms14945) to this surface in the absence or presence of galectin-1. Again, the presence of galectin-1 did not affect the binding of KKO. From this model experiment, we conclude that galectin-1 is unlikely to participate in the immune complex formation that triggers the HIT response. We have added these data as S3_Fig and discussed the data on page 13, line 283.

2. Does galectin-1 induce similar platelet responses in platelet rich plasma (PRP) and whole blood?

3. Can concentrations of gal-1 that induced platelet activation in vitro be reached in human blood under some circumstances?

Thank you for raising these points. First to 3: This is an important point. Previous studies have reported serum and plasma concentrations of galectin-1 in healthy individuals ranging from 1-17 nM. However, given the potential interaction of gal-1 with several cellular components of the vasculature, the measured soluble levels of gal-1 might be an underestimation of the total gal-1 present in a blood vessel (similar counts for the anticoagulant TFPI, the majority is bound to the vessel wall). Concentrations of up to 2.4 µM have been observed within the bovine spleen (doi: 10.1093/oxfordjournals.jbchem.a021493), indicating that local concentrations might be orders of magnitude above those in plasma and might even increase further during inflammation. The concentrations that we have used in our experiments are in the range of other previous studies (e.g. references 9 and 10 in manuscript) and is around the monomer-dimer equilibrium constant of gal-1 (7µM, doi: 10.1021/bi961181d). Studies in galectin-1-deficient mice showed altered platelet responses e.g. in a model of bleeding time. This suggests that galectin-1 can reach sufficiently high concentrations and can be active in a plasma environment. Since gal-1 and CXCL4 are present in platelets, they might act in concert at sites of vascular injury. To 2: We have performed the proposed experiment and found that unlike collagen, galectin-1 does not induce platelet aggregation in plasma (at least up to 6 µM). These findings were added as S2_Fig in the revised manuscript. Plasma contains several abundant proteins that might sequester galectin-1 (IgM, a2-macroglobulin, fetuin). Plasma may also contain an unspecified interactor that competes with the carbohydrate receptors for galectin-1 on platelets.

We have further discussed our findings and the obvious discrepancies in the data on page 15, line 338.

4. In the discussion interaction of gal-3 with GPVI on platelets is mentioned. Could a similar mechanism be responsible for galectin-1 mediated effects? Authors should check how blocking of GPVI or co-stimulation with GPVI-ligands (collagen or CRP) affect galectin-1 mediated platelet activation and aggregation.

We agree with the reviewer that this would indeed be an interesting possibility to pursue, in the light of the heavy glycosylation of the platelet GPVI or GPIb (reviewer #1). However, unlike gal-1, gal-3 has a collagen-like domain on which the interaction with GPVI is likely based (doi: 10.1182/blood.2019002649). Our findings with the GPVI agonists collagen and CRP-XL indicate that the pattern of platelet activation is different (e.g. robust platelet aggregation and P-selectin expression induced by GPVI stimulation), which would speak against GPVI as a bona-fide receptor for gal-1. Moreover, co-stimulation with GPVI ligands and gal-1 would add an additional layer of complexity to our study, taking it outside of its current scope. Also in response to reviewer #1, we have discussed this possibility on starting on page 17, line 420.

5. Galectin-1 induced CXCL4 release and surface expression, however (according to results shown in figure 2) this was not able to induce the effects exerted by exogenously added CXCL4. Does endogenous CXCL4 play a role for galectin-1 mediated effects? Given the fact, that platelet-derived CXCL4 would not be sufficient for the observed additive effects with galectin-1, what would be the main source of such concentrations of exogenous CXCL4 in vivo?

We thank the reviewer for raising this point. We were also aware of this and we have considered neutralizing CXCL4 in the presence of gal-1. From our own experience, CXCL4 is difficult to neutralize using antibodies. In fact, this might even result in the formation of immune complexes that activate platelets (Greinacher et al., doi: 10.1038/ncomms14945). Thus, we have chosen heparin as a CXCL4 neutralizing agent, since several studies including our own have shown that heparin is able to block CXCL4 function (Vajen et al., doi: 10.1080/20013078.2017.1322454.). We now show results of experiments in which we added heparin to our experimental setup. The presence of heparin appeared to somewhat sensitize the platelets, yet there were no differences in the course of platelet activation by gal-1. We have now included these data as S1_Fig and these were discussed on page 8, line 179.

6. In figure 2 and 3 a positive control (e.g. a classical platelet activator) should be included for each experiment. This is particularly important for P-Selectin in figure 2, as not only conflicting results to previously reported data have been observed, but results stay also in some conflict with the observations that galectin-1 induced CXCL4-release which is also supposed to be stored in alpha granules. The authors might consider to validate the results using a second P-Selectin antibody.

We have now added an appropriate positive control (collagen or collagen-related peptide) to the Figures 1B, 2, and 3. The reviewer is correct that CXCL4 and P-selectin are stored in a-granules. As outlined on page 14, line 328 of the manuscript, several studies have shown that a-granule content can be differentially released (e.g. Battinelli et al., doi: 10.1182/blood-2011-02-334524). We did see robust P-selectin presentation when platelets were triggered with collagen (e.g. Figure 2F, 3F), so we believe that our routinely used anti-P-selectin antibody gives reliable results.

7. To investigate gal-1 binding to platelet integrin-receptor after neuraminidase treatment authors should make additional experiments using antibody-based integrin-inhibitors (e.g. Abciximab) rather than only small molecule inhibitor Tirofiban.

This valid point has also been raised by reviewer #1. Following this suggestion, we have expanded figure 7 with experimental data implementing both eptifibatide and Reopro. We have discussed the observations on page 12 line 270 and page 17, line 408.

8. Authors should discuss in more detail why neuraminidase abolishes CXCL4-induced platelet aggregation but on the other side lowered threshold for P-Selectin expression.

This is indeed an interesting and well observed point. We believe that neuraminidase treatment actually pre-activates platelets, thereby also lowering their activation threshold, as was also shown by others (doi: 10.1128/iai.00213-18 and 10.3324/haematol.2019.215830). This is particularly evident with the platelet activation markers measured by FACS, which is a particularly sensitive method. We do not consider this a CXCL4-specific effect. Since it is yet unknown what molecules actually mediate binding of CXCL4 to platelets, we can only speculate about the mechanistic background of the observation that neuraminidase treatment reduces platelet aggregation. Since CXCL4 is a much weaker platelet stimulant than e.g. collagen, and glycosaminoglycans mediate binding to CXCL4, it might be a rather unspecific effect. We have discussed this on page 17 line 402.

Minor comments:

1. In figure 1 one could assume some co-localization of galectin-1 and CXCL4, however the cross-section used for intensity profile does not seem representative for the figure. Increased image quality would strengthen the results, additionally for the PF4-antibody a respective isotype control image should be demonstrated to show antibody specificity.

In the revised figure 1, we have added a second image of an independent experiment that shows co-localization of gal-1 and CXCL4. We unfortunately do not have an isotype control to our polyclonal anti-CXCL4 antibody, and an isotype control is actually not going to prove specificity of the antibody (for this we would need human platelets deficient in CXCL4). Alternatively, we provide an image (reviewer only figure 1) of the endothelial cell line EA.Hy926, which do not express CXCL4, treated without or with purified CXCL4 and subsequently stained with the anti-CXCL4 antibody (Peprotech #500-P05, the affinity-purified rabbit polyclonal routinely used by us and by other expert groups e.g. Kowalska and Poncz, doi: 10.1111/jth.13069) and with Alexa Fluor (AF) 647-coupled anti-rabbit secondary antibody (the same procedure as the platelet staining). It is visible that neither the primary nor the secondary antibodies stain other components/compartments of the cells, apart from the exogenously added CXCL4.

Please see the uploaded point-by-point reply file for the figure.

2. The experimental design in figure 4 is not appropriate to investigate influence of CXCL4 on galectin-1-induced integrin activation. Further effects of increasing concentrations of CXCL4 cannot be excluded as 3 µM of galectin-1 already showed almost maximal activation. Experiment should be repeated using a lower dose of galectin-1 or the conclusion and interpretation must be revised.

As suggested, we have now replaced figure 4 with 1µM of galectin-1, a concentration at which the effects of CXCL4 are more appreciable. We have discussed the data starting on page 9, line192. 

3. Result section related to figure 4 (page 8, line 150-160) is confusing and somewhat contradictory.

As a consequence of the replacement of Figure 4, the section has been rephrased.

4. It is not always clear whether galectin-1, CXCL4 and neuraminidase were added simultaneously or which treatment was first.

Platelets were treated with neuraminidase first and subsequently stimulated with the other factors. When gal-1 and CXCL4 were combined, they were pre-incubated together prior to addition to the platelets. This is now clarified at the respective sections.

5. Y-axis in graphs showing PAC-1 expression should be labeled “activated integrin expression”.

This has been changed as suggested.

---

## [Decision Letter · Decision Letter 1]

16 Dec 2020

Galectin-1 and platelet factor 4 (CXCL4) induce complementary platelet responses.

PONE-D-20-16580R1

Dear Dr. Koenen,

We’re pleased to inform you that your manuscript has been judged scientifically suitable for publication and will be formally accepted for publication once it meets all outstanding technical requirements.

Within one week, you’ll receive an e-mail detailing the required amendments. During this process, please also address the two remaining points raised by reviewer 2. When these have been addressed, you’ll receive a formal acceptance letter and your manuscript will be scheduled for publication.

Kind regards,

Christian Schulz

Academic Editor

PLOS ONE

Additional Editor Comments (optional):

Reviewers' comments:

Reviewer's Responses to Questions

**Comments to the Author**

1. If the authors have adequately addressed your comments raised in a previous round of review and you feel that this manuscript is now acceptable for publication, you may indicate that here to bypass the “Comments to the Author” section, enter your conflict of interest statement in the “Confidential to Editor” section, and submit your "Accept" recommendation.

Reviewer #1: All comments have been addressed

Reviewer #2: (No Response)

2. Is the manuscript technically sound, and do the data support the conclusions?

Reviewer #1: Yes

Reviewer #2: Partly

3. Has the statistical analysis been performed appropriately and rigorously? 

Reviewer #1: Yes

Reviewer #2: Yes

4. Have the authors made all data underlying the findings in their manuscript fully available?

Reviewer #1: Yes

Reviewer #2: Yes

5. Is the manuscript presented in an intelligible fashion and written in standard English?

Reviewer #1: Yes

Reviewer #2: Yes

6. Review Comments to the Author

Reviewer #1: The authors have significantly improved the paper and have well addressed the issues of concerns and answered the open questions partly by performing new experiment, so that I recommend strongly acceptance in the present form.

Reviewer #2: I acknowledge that the authors put considerable effort to answer and discuss the reviewers’ questions, and therefore also performed some interesting additional experiments such as SPR (to investigate a possible influence on HIT). The additional results and technical controls support and endorse some of the data, however they do not increase evidence for a possible and conclusive role of the findings in an in vivo situation. Further, (even after extensive discussion by the authors) the findings about “interaction” of CXCL4 with Neuraminidase/Neuraminidase-desialylated platelets remain somewhat elusive/confusing. Yet, and despite the lack of an obvious translational relevance, after this careful and honest revision the study considerably improved, and the data may build the groundwork for further studies investigating the role of gal-1/CXCL4-interaction in disease models of thrombo-inflammation (for this genetic animal models probably will be required).

I have 2 more comments that should be addressed:

1. New data now presented in fig. S1 suggest that platelet-derived/released CXCL4 induced by gal-1 does NOT contribute to (auto-/paracrine) platelet activation as compared to exogenously added CXCL4 (which induces robust P-Selectin expression, see fig. 3).

Thus, the conclusion made by the authors “… activation of platelets by gal-1 led to an increased release of CXCL4, which not only co-localized with gal-1 on the platelet surface, but was also able to activate platelets by itself.” (revised manuscript line 313-315) is not supported by this data and must be revised.

2. In vivo platelets are considered to be the main source of CXCL4. Given that according to the newly provided data in fig. S1 platelet-derived CXCL4 does NOT contribute to activating effects of CXCL4, the authors should discuss the alternative source of CXCL4 (rather than platelets) in vivo (see my comment 5 of the first review). Otherwise the setting chosen by the authors (i.e. adding exogenously rather high doses of CXCL4 in vitro) seems very artificial and constitutes a major limitation for translational interpretation of the findings.

Therefore, I suggest to add this information in the title and change it accordingly to: “Galectin-1 and platelet factor 4 (CXCL4) induce complementary platelet responses IN VITRO”

7. PLOS authors have the option to publish the peer review history of their article (what does this mean?). If published, this will include your full peer review and any attached files.

Reviewer #1: No

Reviewer #2: No

---

## [Editor Report · Acceptance letter]

22 Dec 2020

PONE-D-20-16580R1 

Galectin-1 and platelet factor 4 (CXCL4) induce complementary platelet responses in vitro. 

Dear Dr. Koenen:

I'm pleased to inform you that your manuscript has been deemed suitable for publication in PLOS ONE. Congratulations! Your manuscript is now with our production department. 

Kind regards, 

on behalf of

Prof. Christian Schulz 

Academic Editor

PLOS ONE